# Preoperative Intensified Chemoradiation with Intensity-Modulated Radiotherapy and Simultaneous Integrated Boost Combined with Capecitabine in Locally Advanced Rectal Cancer: Long-Term Outcomes of a Real-Life Multicenter Study

**DOI:** 10.3390/cancers15235702

**Published:** 2023-12-04

**Authors:** Marco Lupattelli, Elisa Palazzari, Jerry Polesel, Giuditta Chiloiro, Ilaria Angelicone, Valeria Panni, Luciana Caravatta, Saide Di Biase, Gabriella Macchia, Rita Marina Niespolo, Pierfrancesco Franco, Valeria Epifani, Elisa Meldolesi, Flavia de Giacomo, Marco Lucarelli, Giampaolo Montesi, Giovanna Mantello, Roberto Innocente, Mattia Falchetto Osti, Maria Antonietta Gambacorta, Cynthia Aristei, Antonino De Paoli

**Affiliations:** 1Radiation Oncology Section, Department of Medicine and Surgery, University of Perugia and Perugia General Hospital, 06129 Perugia, Italy; marco.lupattelli@ospedale.perugia.it (M.L.); valeria.epifani@ospedale.perugia.it (V.E.); cynthia.aristei@unipg.it (C.A.); 2Radiation Oncology Department, Centro di Riferimento Oncologico di Aviano (CRO) IRCCS, 33081 Aviano, Italy; roberto.innocente@cro.it (R.I.); adepaoli@cro.it (A.D.P.); 3Unit of Cancer Epidemiology, Centro di Riferimento Oncologico di Aviano (CRO) IRCCS, 33081 Aviano, Italy; polesel@cro.it; 4Department of Diagnostic Imaging, Radiation Oncology and Hematology, Fondazione Policlinico Universitario A. Gemelli—IRCCS, Università Cattolica del Sacro Cuore, 00168 Roma, Italy; giuditta.chiloiro@policlinicogemelli.it (G.C.); elisa.meldolesi@policlinicogemelli.it (E.M.); mariaantonietta.gambacorta@policlinicogemelli.it (M.A.G.); 5Department of Radiation Oncology, Sant’Andrea Hospital-Sapienza University of Rome, 00189 Rome, Italy; ilaria.angelicone@uniroma1.it (I.A.); flavia.degiacomo@uniroma1.it (F.d.G.); mattiafalchetto.osti@uniroma1.it (M.F.O.); 6Department of Radiation Oncology, Azienda Ospedaliero Universitaria delle Marche, 60002 Ancona, Italy; valeria.panni@ospedaliriuniti.marche.it (V.P.); giovanna.mantello@ospedaliriuniti.marche.it (G.M.); 7Radiation Oncology Department, SS Annunziata Hospital, G.D’Annunzio University of Chieti-Pescara, 66100 Chieti, Italy; lcaravatta@hotmail.com (L.C.); lucarellimarco1991@libero.it (M.L.); 8Radiation Oncology Unit, Santa Maria della Misericordia Hospital, 45100 Rovigo, Italy; giampaolo.montesi@aulss5.veneto.it (G.M.); saide.dibiase@aulss5.veneto.it (S.D.B.); 9Radiation Oncology Unit, Responsible Research Hospital, 86100 Campobasso, Italy; 10Radiation Oncology Unit, Fondazione IRCCS San Gerardo dei Tintori, 20900 Monza, Italy; ritamarinaniespolo@gmail.com; 11Department of Translational Medicine, University of Eastern Piedmont, 28100 Novara, Italy; pierfrancesco.franco@uniupo.it; 12Department of Radiation Oncology, “Maggiore della Carità” University Hospital, 28100 Novara, Italy

**Keywords:** rectal cancer, preoperative chemoradiotherapy (CRT), intensity modulated RT (IMRT), simultaneous integrated boost (SIB), real-life clinical practice, radiotherapy dose, long-term results

## Abstract

**Simple Summary:**

Preoperative chemoradiation (CRT) for rectal cancer with the intensification of radiotherapy (RT) using dose escalation to the tumor volume has been shown effective in improving tumor regression with high compliance to treatment and low toxicity rates. Most dose-escalation trials used conventional 3D conformal RT with concurrent capecitabine. More recently, phase I-II trials investigated intensified RT programs with advanced intensity-modulated RT and simultaneous boost (IMRT-SIB) supported by image-guided RT (IGRT) techniques, highlighting their feasibility and promising activity. However, only limited data on long-term outcomes are available. We analyzed the long-term results of a retrospective, multicenter experience with preoperative capecitabine-based CRT intensification with IMRT-SIB in real-life clinical practice in 10 Italian institutions. The use of moderate IMRT-SIB dose intensification with a full dose of concurrent capecitabine was safe and well tolerated. In addition, an organ preservation strategy has been shown feasible in carefully selected, responsive patients with a promising long-term rectal preservation rate. Long-term local control, progression-free and overall survival rates compared favorably with conventional CRT trials. Given the higher incidence of distant metastases in the subset of high-risk patients, the incorporation of IMRT-SIB and capecitabine with a more effective systemic therapy component may represent a new area of investigational interest while the use of IMRT-SIB and capecitabine in a primarily organ-preservation strategy may be a valuable option for low-intermediate-risk patients.

**Abstract:**

Background: Despite the feasibility and promising activity data on intensity-modulated RT and simultaneous integrated boost (IMRT-SIB) dose escalation in preoperative chemoradiation (CRT) for locally advanced rectal cancer (LARC), few data are currently available on long-term outcomes. Patients and Methods: A cohort of 288 LARC patients with cT3-T4, cN0-2, cM0 treated with IMRT-SIB and capecitabine from March 2013 to December 2019, followed by a total mesorectal excision (TME) or an organ-preserving strategy, was collected from a prospective database of 10 Italian institutions. A dose of 45 Gy in 25 fractions was prescribed to the tumor and elective nodes, while the SIB dose was prescribed according to the clinical practice of each institution on the gross tumor volume (GTV). Concurrent capecitabine was administered at a dose of 825 mg/m^2^ twice daily, 7 days a week. The primary objective of the study was to evaluate long-term outcomes in terms of local control (LC), progression-free survival (PFS) and overall survival (OS). The secondary objective was to confirm the previously reported feasibility, safety and efficacy (pCR, TRG1-2 and downstaging rates) of the treatment in a larger patient population. Results: All patients received a dose of 45 Gy to the tumor and elective nodes, while the SIB dose ranged from 52.5 Gy to 57.5 Gy (median 55 Gy). Acute gastrointestinal and hematologic toxicity rates of grade 3–4 were 5.7% and 1.8%, respectively. At preoperative restaging, 36 patients (12.5%) with complete or major clinical responses (cCR or mCR) were offered an organ-preserving approach with local excision (29 patients) or a watch and wait strategy (7 patients). The complete pathologic response rate (pCR) in radically operated patients was 25.8%. In addition, 4 TME patients had pT0N1 and 19 LE patients had pT0Nx, corresponding to an overall pT0 rate of 31.3%. Of the 36 patients selected for organ preservation, 7 (19.5%) required the completion of TME due to unfavorable pathologic features after LE or tumor regrowth during W-W resulting in long-term rectal preservation in 29 of 288 (10.1%) of the total patient population. Major postoperative complications occurred in 14.2% of all operated patients. At a median follow-up of 50 months, the 5-year PFS and OS rates were 72.3% (95% CI: 66.3–77.4) and 85.9% (95% CI: 80.2–90.1), respectively. The 5-year local recurrence (LR) rate was 9.2% (95% CI: 6.0–13.2), while the distant metastasis (DM) rate was 21.3% (95% CI: 16.5–26.5). The DM rate was 24.5% in the high-risk subset compared to 16.2% in the low-intermediate risk group (*p* = 0.062) with similar LR rates (10% and 8%, respectively). On multivariable analysis, cT4 and TRG3–5 were significantly associated with worse PFS, OS and metastasis-free survival. Conclusions: Preoperative IMRT-SIB with the moderate dose intensification of 52.5–57.5 Gy (median 55 Gy) and the full dose of concurrent capecitabine confirmed to be feasible and effective in our real-life clinical practice. Organ preservation was shown to be feasible in carefully selected, responsive patients. The favorable long-term survival rates highlight the efficacy of this intensified treatment program. The incorporation of IMRT-SIB with a more effective systemic therapy component in high-risk patients could represent a new area of investigational interest.

## 1. Introduction

Preoperative chemoradiotherapy (CRT) or short course radiotherapy followed by total mesorectal excision (TME) is the standard treatment for patients (pts) with locally advanced rectal cancer (LARC). When “downsizing” and “downstaging” are required (cT3-4, N0-2, mesorectal fascia involvement [MRF] or cT3, MRF +/− N0 of the lower rectum), CRT is recommended. Although this approach leads to an improvement in local control and a significantly higher rate of complete pathologic recurrence (pCR), which is between 13% and 20%, about 25–30% of these patients develop distant metastases. Adjuvant chemotherapy has not provided a survival benefit; moreover, this treatment approach has been associated with suboptimal compliance and relevant toxicity [1]. Preoperative intensification of CRT has been an area of major investigational interest in LARC over the past two decades. The favorable outcomes in tumor response the after treatment of patients with pCR [2,3] and the chance for organ preservation in selected patients with complete clinical response (cCR) [4,5,6,7,8] have prompted clinical researchers to intensify preoperative CRT, mainly by adding the second drug oxaliplatin to the standard fluoropyrimidine-based CRT. However, the benefit of adding oxaliplatin reported in phase II trials [9,10,11] was not confirmed in phase III trials comparing the combination of concurrent oxaliplatin and 5-fluorouracil (FU) or capecitabine with RT 50 Gy to the standard therapy of FU/capecitabine and 50 Gy followed by adjuvant FU or capecitabine [12,13,14], or capecitabine +/− oxaliplatin [15,16]. Only one study reported a benefit in terms of pCR rate and 3-year disease-free survival (DFS) [15]. Most studies failed to demonstrate an improvement in oncologic outcomes and reported a significant increase in toxicity rates. Interestingly, two different strategies for intensifying preoperative CRT were investigated in the INTERACT study. The addition of oxaliplatin to standard capecitabine-based CRT (intensified CT component) was compared with the intensification of the radiotherapy component by 3D-conformal RT (3D-CRT), in which the dose was increased to the tumor volume (concomitant-boost (CB), to achieve a total dose of 55 Gy over 5 weeks (10 Gy/10 fractions, twice weekly, in addition to the 45 Gy for the mesorectum and elective nodes) and concurrent capecitabine 825 mg/m^2^ twice daily, 7 days/week. Adjuvant FU- or capecitabine-based CT was recommended for non-complete responsive patients and/or whose nodes were positive at surgery. Although no difference was found in terms of pCR (pT0N0), local control and survival rates, the study showed a significant difference in major pathological tumor response (tumor regression grade-TRG1-2) [17], better compliance to treatment and lower toxicity rates in favor of intensification of RT and concurrent capecitabine [18]. A dose–response relationship for tumor regression after preoperative CRT was also reported in a dose–response model by Appelt et al. and a meta-analysis of phase I-II trials confirmed these promising observations [19,20].

Based on these data, innovative radiation dose escalation programs using modern RT techniques such as intensity-modulated RT (IMRT) with simultaneous integrated boost (SIB) in combination with image-guided RT (IGRT) have been tested in preoperative CRT for treatment intensification. A recent systematic review and meta-analysis of phase II studies on dose escalation with IMRT-SIB, albeit using different radiation doses and SIB modalities, indicates the feasibility and promising efficacy of modern RT techniques for moderate dose escalation of 54–60 Gy. However, only limited data on long-term outcomes and late toxicity are available [21]. In our previous retrospective multicenter real-life study, we reported encouraging short-term results with preoperative IMRT-SIB in the dose range of 52.5–57.5 Gy (median 54 Gy) in 25 fractions and concurrent capecitabine at 825 mg/m^2^ twice daily, 7 days/week, in a group of 76 LARC patients [22]. This intensified approach resulted in limited acute toxicity (6.6% grade ≥3 gastrointestinal (GI) and 10.5% grade ≥3 overall toxicity), high compliance to treatment (97% and 84% for the RT and CT components, respectively) and encouraging response rates (22% pCR and 27.8% pT0-TRG1 rates). In the current study, we report the long-term results of this multicenter expanded cohort of patients with LARC treated with the same preoperative intensified CRT program [22].

## 2. Material and Methods

### 2.1. Study Design and Objectives

This is a real-life retrospective study of a cohort of 288 patients with LARC collected in a prospective database of 10 Italian institutions and treated with an intensified preoperative CRT program with IMRT-SIB and concurrent capecitabine between March 2013 and December 2019. The study also includes the 76 patients from our previous report (22). The primary objective was to evaluate long-term outcomes in terms of local control (LC), progression-free (PFS) and overall survival (OS). The secondary objective was to confirm the previously reported feasibility, safety and efficacy (pCR, TRG1-2 and downstaging rates) of the treatment in a larger patient population. The study was approved by the independent Ethics Committee (EC) of the CRO-IRCCS in Aviano (Italy) and accepted by all participating institutions.

### 2.2. Eligibility Criteria

The inclusion criteria were: histologic diagnosis of adenocarcinoma of no metastatic extraperitoneal LARC who underwent neoadjuvant CRT with IMRT-SIB and concurrent capecitabine (or 5-fluorouracil as continuous infusion) followed by TME surgery, or organ-preserving strategies in selected patients who responded completely, according to the current clinical practice of each participating institution. A staging and restaging MRI of the pelvis was required, and a follow-up of at least 2 years was mandatory. Data were collected in regard to clinical history, physical and digital rectal examination (DRE), blood and chemistry profile with CEA determination, and staging examinations including colonoscopy with biopsy, CT scan of chest and abdomen, endoscopic ultrasound, and pelvic MRI.

### 2.3. Preoperative Chemoradiation

All patients underwent preoperative CRT with IMRT with SIB and concurrent capecitabine. Elective volumes (CTV2) were defined according to the international consensus guidelines for the delineation of target volumes [23]. The boost clinical target volume (GTV) was defined based on MRI and/or PET-CT for both the rectal tumor and any positive extramesorectal lymph nodes. The boost planning target volume (PTV) takes into account organ motion and set-up margins, according to the center definition. The bowel bag, bladder, vagina, anal sphincter and femoral heads were considered organs at risk (OARs) [24]. Patients received a dose of 45 Gy in 25 fractions to the tumor and elective nodes, while the SIB dose ranged from 52.5 Gy to 57.5 Gy (median 55 Gy). A cone-beam CT scan (or portal vision if IGRT was not available) to verify position was performed according to institutional protocols. Concurrent CT consisted of oral capecitabine at a dose of 825 mg/m^2^ twice daily, 7 days per week, throughout the entire course of treatment.

Clinical response was assessed at restaging with pelvic MRI, DRE, endoscopy and endorectal ultrasound (EUS), if available, planned at 6 to 8 weeks after the end of CRT.

Complete clinical response (cCR) was defined as no residual mass on DRE with a fully normalized rectal wall on endoscopy, no diffusion restriction on the rectal wall and no residual nodes or only nodes ≤ 5 mm (cN0) on diffusion-weighted MRI (DWI). Major clinical response (mCR) was defined as no residual mass on DRE, the presence of a small mucosal irregularity or superficial ulcer ≤ 2 cm in diameter on endoscopy, no clear areas of residual hyperintense signals on the rectal wall, and no residual nodes or only nodes ≤ 5 mm (cN0) in size on DWI [25,26]. Acute toxicity was assessed weekly during treatment and at restaging prior to surgery and scored according to CTCAE v. 4.0.

### 2.4. Surgery

Surgery was planned 10–12 weeks after CRT. Total mesorectal excision or partial mesorectal excision (PME) was performed according to the original tumor location and/or amount of residual disease at restaging. The standardized TME technique included low anterior resection (LAR), abdominal perineal resection (APR) or the Hartmann procedure (if necessary). Temporary ileostomy or colostomy was at the discretion of the treating surgeon. In carefully selected patients with cCR or mCR, an organ preservation strategy was offered, mainly in the context of investigational trials [25,27], and included a watch-and-wait (W-W) approach or transanal local excision (LE) procedures, depending on clinical response and institutional experience. Pathological response was scored according to the standardized five-point tumor regression grade (TRG) of Mandard et al. [17]. A pCR was defined as no visible microscopic disease in the primary tumor (TRG1) and in the lymph nodes (pT0N0). Patients with pT0Nx after LE were considered complete responders. Postoperative complications assessed within 30 days after surgery were graded according to the Clavien-Dindo classification [28].

### 2.5. Adjuvant Chemotherapy

Adjuvant CT was usually given to patients who did not respond to therapy (TRG3-5) or whose nodes were positive at surgery, according to the guidelines of each participating center, and consisted mainly of oral capecitabine 1000 mg/m^2^ twice daily, days 1 to 14, for 6 cycles. FU/capecitabine-based CT with or without a second drug was also an optional regimen.

### 2.6. Follow-Up and Monitoring of Late Toxicity

Patients were evaluated every three months for the first two years, every six months from the third to the fifth year and annually thereafter. For patients who underwent an organ preservation strategy with LE or W-W option, a special, more intensive follow-up was provided. Late toxicity was assessed at each clinical evaluation and scored according to the CTCAE criteria, version 4.0.

### 2.7. Statistical Analysis

Qualitative variables were expressed as percentages and quantitative variables as median and interquartile range (Q1–Q3). The odds ratio (OR) and 95% confidence interval (CI) for TRG1-2 versus TRG3-5 were calculated using an unconditional logistic regression model adjusting for sex, age, clinical T and N status, MRF involvement, distance of the lower tumor pole from the anal verge, RT dose levels, and time between end of CRT and surgery.

For each patient, the time at risk was calculated from surgery to the event of interest, death or last follow-up, whichever occurred first. For patients selected for W-W, the time at risk was calculated from restaging after preoperative CRT. The event of interest was defined as local recurrence, defined as R2 resection of the primary tumor or recurrence in the primary tumor bed after R0–R1 resection, for local recurrence-free survival; distant metastases (at any site) for distant metastasis-free survival; death for OS; local recurrence, distant metastases or death for PFS. Patients who experienced local recurrence after local excision or who had tumor regrowth during a W-W option and underwent successful surgical salvage were considered free of local recurrence. Patients who refused salvage TME were censored as having persistent disease for the analysis of PFS and local control. The Kaplan–Meier method was used to calculate the 5-year OS and PFS [29]. To account for competing risk, the cumulative incidence of local failure and distant metastases was evaluated by the cumulative incidence [30 ≥ 55 Gy], and differences by blood parameters were tested by the Gray test [30]. Hazard ratios (HR) and corresponding 95% CIs were calculated using multivariable Cox proportional hazards models [29]. For local failure and distant metastasis, risk estimates were adjusted for competing risk according to the Fine–Gray model [31]. Statistical analyses were performed using SAS 9.4 and R 4.1. Statistical significance was claimed for *p* < 0.05.

## 3. Results

A total of 288 patients were included in this analysis between March 2013 and December 2019. In most pts (53.8%), the tumor was located in the distal rectum with a median distance to the anal verge of 50 mm (range 30–75). Most of them had unfavorable stage III disease, of which 31.6% were in the cN2 subgroup and 18.8% had a cT4 component. In addition, mesorectal fascia involvement (MRF) was found in 48.3% of cases. Therefore, a large proportion of patients (61.1%) had high-risk disease at presentation [32]. The patient and tumor characteristics are listed in Table 1.

All patients received the prescribed dose of 45 Gy, and the SIB dose ranging from 52.5 Gy to 57.5 Gy in 25 fractions, corresponding to an equivalent dose in 2 Gy-fraction (EQD2) of 53.24 Gy, 55.22 Gy, 56.55 Gy, 58 Gy and 59.94 Gy, respectively (α/β = 5.06 Gy for rectal tumors) [33]. The median IMRT-SIB dose was 55 Gy/25 fractions (EQD2 = 56.55 Gy). IMRT-SIB was performed in 18%, 28%, 48% and 6% of patients using the “step and shoot”, “sliding window”, VMAT and Tomotherapy techniques, respectively.

Overall, 99.3% of patients completed the planned IMRT-SIB therapy and 78.1% received the prescribed capecitabine dose. The remaining patients interrupted CT for an average of 10 days (range 7–12) due to hematologic and/or gastrointestinal toxicity or other causes and resumed therapy with a dose modification after recovery. Thus, optimal compliance with preoperative CRT was observed in at least 78% of patients. In general, the treatment was well tolerated. The most common grade 3–4 acute toxicity was gastrointestinal (diarrhea and/or proctitis) and hematologic (leukopenia) and was seen in 5.9% and 2.1% of patients, respectively. Grade 2 toxicity with moderate diarrhea/proctitis, cystitis and/or protracted moderate leukopenia was reported in 22% of cases. No grade 5 toxicity was reported.

At preoperative restaging, performed after a median of 9 weeks (6–12) from the end of CRT, 36 of the 288 (12.5%) patients achieved cCR (6.9%) or mCR (5.6%). The preoperative CRT programs, clinical response, acute toxicity and treatment compliance are shown in Table 2.

Surgery was performed a median of 12 weeks (range 10–15) from the end of CRT and consisted of a TME in 252 patients (87.5%), while the 36 patients with cCR or mCR were offered an organ-preserving approach. Of these 36 cases, 29 (80.5%) received an LE procedure; the other 7 patients (19.5%) were treated with a non-operative management with a W-W strategy.

Overall, 65 of 252 (25.8%) radically operated patients achieved a pCR (pT0N0) and 4 additional TME patients had a pT0N1 stage. In addition, 19 out of 29 (65.5%) patients who underwent LE were found to have a pT0Nx stage. Thus, 88 patients (31.3%) received a pT0-TRG1. Interestingly, 100% of patients with cCR and 75% of patients with mCR who underwent LE were found to have a pT0-1 (TRG1-2) stage. Six of these LE patients required the completion TME surgery because of TRG > 2 (2 pT1 patients) or stage pT2-3 disease (4 patients). Of the 7 cCR patients who were offered a W-W option, 2 had tumor regrowth and underwent radical surgery (1 patient) or LE because they declined TME (1 patient). Thus, 7 of 36 patients (19.5%) initially selected for an organ-preserving approach had to undergo TME surgery because of unfavorable histologic features after LE or tumor regrowth during W-W, resulting in 29 of 36 (80.5%) of the selected patients, 29 of 288 (10.1%) of the total patient population, undergoing rectal preservation. Details of the concordance between clinical and pathologic T-stage in patients who underwent LE are shown in Table 3.

The major pathological response rate (TRG1-2) was achieved in 177 of the 267 available patients (66.3%). Overall, tumor downstaging from cT3-4 to pT0-2 was achieved in 170 of 281 (60.5%) of all patients undergoing surgery (TME and LE) and nodal downstaging from cN1-2 to pN0 was achieved in 208 of 252 (82.5%) of patients undergoing TME.

Major postoperative complications (Clavien–Dindo grade ≥ 3) within 30 days of surgery occurred in 41 of 289 patients (14.2%), including 8 patients who required a completion radical TME surgery after LE or tumor regrowth during the W-W option. Major postoperative complications after LE occurred in 1 of 28 patients (3.5%) who had rectal bleeding that required endoscopic treatment and blood transfusions. Five other LE patients (17.8%) had minor complications, including dehiscence of the rectal suture and persistent pain, fully recovered. No postoperative mortality was reported (Table 4).

Overall, 77 out of 288 (26.7%) patients received adjuvant chemotherapy. The indication for therapy was mainly based on the pathological stage after preoperative CRT and surgery rather than clinical stage at initial presentation. Most of these patients were at stage pT3-4 and/or pN1-2 of the disease, while the indication for stage pT2-3N0 was mainly based on TRG ≥ 3. No patient with pCR received adjuvant therapy. Oral capecitabine 1000 mg/m^2^ twice daily, days 1 to 14, for 6 cycles was the most common adjuvant therapy, while capecitabine or FU with oxaliplatin (Capox or Folfox regimen) was used in a smaller number of patients.

At a median follow-up of 50 months (Q1–Q3: 35–62 months), the 5-year OS and 5-year PFS rates were 85.9% and 72.3%, respectively. The 5-year LR and DM rates were 9.2% (95% CI: 6.0–13.2) and 21.3% (95% CI: 16.5–26.5), respectively. Overall, 215 (74.6%) patients were disease- free at the time of the present analysis, while 31 (10.8%) had died from any cause. Patterns of failure by surgical option are shown in Table 5.

The Kaplan–Meier survival curves of PFS, OS and the cumulative incidence of LR and DM are reported in Figure 1.

The 5-year OS and PFS rates were significantly higher in low and intermediate-risk patients than in high-risk patients: 93.7% versus 81.2% (*p* = 0.026) and 78.6% versus 68.4% (*p* = 0.039), respectively. Interestingly, the 5-year LR rates in the high-risk patients showed a rate of 10.0%, which was well comparable to the 8.0% of low and intermediate-risk patients, while a trend, albeit not significant, towards a higher cumulative incidence of DM was found in the high-risk patients (24.5% versus 16.2%, *p* = 0.062). The cumulative incidence of LR and DM in high- and low intermediate-risk patients is shown in Figure 2.

MRF involvement was associated with poor response probability (TRG3–5) compared to TRG1-2 in multivariable analysis (OR 2.0; 1.05–3.79, *p* = 0.035). A favorable trend for better response was found for a higher IMRT-SIB dose ≥ 55 Gy (OR 0.57; 0.32–1.01, *p* = 0.056). These data are shown in Table 6.

The presence of the cT4 stage and TRG3-5 was associated with a significantly worse 5-year OS and PFS rate and DM incidence. The administration of adjuvant chemotherapy was associated with worse PFS (*p* = 0.013), and an unfavorable trend was also observed for DM incidence (*p* = 0.055). The hazard ratios (HR) and corresponding 95% confidence intervals (CIs) for OS, DFS, LR and DM according to the selected factors considered are shown in Table 7 and Table 8.

## 4. Discussion

In this retrospective, multicenter study, we analyzed the experience with preoperative capecitabine-based CRT intensification with modern IMRT-SIB dose escalation in the real-life clinical practice of 10 Italian institutions. The IMRT-SIB program with a dose range of 52.5–57.5 Gy (median 55 Gy) in 25 fractions (2.1–2.3 Gy/fraction) and concurrent administration of capecitabine proved to be feasible and safe in a large population of LARC patients. These results confirm our previous short-term data from the first series of 76 LARC patients who underwent the same treatment program [22]. Our results are comparable to the tumor response and toxicity rates reported in the more favorable dose-escalation arm of 55 Gy with CB 3D-CRT and concurrent capecitabine both in terms of response and compliance (grade ≥ 3 toxicity 6.6–11.5% and pCR 24.4% with pT0-TRG1 32.3%) of the previously mentioned INTERACT trial, although they were obtained in a more favorable patient population (stage T3 and distal T2 only) [18]. Furthermore, our results are also consistent with those reported in the recent meta-analysis of preoperative CRT dose-escalation studies using modern IMRT-SIB techniques and a moderate dose escalation of 54–60 Gy, resulting in a pooled pCR rate of 25.7% and a median toxicity grade ≥ 3 of 9.8% (range 4.6–19.7%) [21]. Although no phase III studies were available for a direct comparison of IMRT dose intensification with standard CRT, it is important to note that the pCR and toxicity data from the meta-analysis and also our pCR and toxicity data compare favorably with the pCR and toxicity rates of the preoperative standard CRT arms with FU or capecitabine and 50 Gy (13–16%) from the previously reported phase III trials [12,13,14,15,16], thus supporting the efficacy of preoperative IMRT dose intensification.

Our IMRT-SIB dose escalation programs ranged from 52.5 Gy to 57.5 Gy. Multivariable analysis found no significant impact on tumor response (pCR, TRG1-2) or downstaging for the different dose levels, although a positive trend was observed for doses ≥ 55 Gy. While these data are meaningful, they should be interpreted with some caution due to the small number of patients considered for each dose level. Despite the wide dose range, most patients (80.3%) received a moderate dose intensification of 54–55 Gy (median 55 Gy). These dose levels are well comparable with the 55 Gy of dose escalation with CB 3D-CRT in the INTERACT trial and with the 54–60 Gy, more commonly 55 Gy, reported in the meta-analysis of the IMRT-SIB studies [21]. Based on the more favorable acute toxicity of grade ≥3 and the pCR rates reported in their meta-analysis compared to conventional CRT series (control arms of randomized trials) [12,13,14,15,16], Hearn et al. [21] concluded that a moderate dose escalation of 55 Gy in 25 fractions can be recommended when IMRT-SIB and concurrent FU or capecitabine are planned for LARC. Remarkably, our results are consistent with the available response rate, compliance to treatment and toxicity data reported in the meta-analysis of modern trials using IMRT-SIB dose escalation, suggesting their reproducibility in real-life clinical practice. As our cohort included a substantial proportion of high-risk patients with locally advanced disease, the reported pCR, major tumor response (TRG1-2) and downstaging rates highlight the efficacy of our intensified preoperative CRT treatment.

Tumor response to preoperative IMRT-SIB and capecitabine had a favorable impact on surgery. The majority of patients with pre-treatment MRF involvement who underwent TME had a negative CRM at the pathologic examination; a positive CRM was found in 5.5% of operated patients. In addition, 74.2% of patients who underwent TME underwent sphincter-preserving surgery with LAR. These data are comparable to those reported in preoperative CRT trials using conventional RT doses or CB 3D-CRT dose escalation (range CRM + 2–11%, sphincter-preserving surgery 70–75%) [14,17,31]. Importantly, in our multicenter experience, 36 of 288 (12.5%) patients who had cCR or mCR at restaging were selected for an organ preservation strategy with LE or a W-W approach. Although a portion of these patients were enrolled in phase II organ preservation trials ongoing at some institutions [5,27], some patients were selected based on individual preference, surgeon attitude and the institution’s multidisciplinary team experience. These data highlight the evolving experience in the assessment of cCR and mCR and the growing interest in the organ preservation approach outside of clinical trials at several institutions. Although we are not able to evaluate the actual impact of our IMRT-SIB and concurrent capecitabine on organ preservation, as this is a retrospective study with potential selection bias, the 9.34% rate of rectal preservation reported in our multicenter study is remarkable. This data compares well with the 10.3% rate reported in the INTERACT trial, which included a prospective organ preservation option for good responsive patients [18]. Despite the small number of patients available, the rates of pT0–1Nx reported after LE (25 of 29 patients = 86.2) and the sustained cCR for patients in W-W (5 of 7 = 71.4%) appear to be consistent with recently published data from the prospective, multicenter ReSARCh observational study by Marchegiani et al. [27], which confirmed the results of previous organ preservation trials [5,6,7,34]. Importantly, our retrospective study shows that these results are also reproducible in real-life clinical practice outside clinical trials when appropriate patient selection is performed [35]. More recently, Garcia-Aguillar et al. [36] reported rectal preservation in almost half of the patients treated with total neoadjuvant therapy (TNT) and selective organ preservation in a large series of 324 patients from a collaborative, phase II trial. Our rectal preservation rate of 9.34% is clearly inferior when compared to these results. However, the efficacy and especially the good tolerability of our intensified IMRT-SIB and capecitabine treatment appear to be of great interest and we consider the reported rectum preservation in current clinical practice to be promising.

The rate of major postoperative complications of 14.2% is consistent with the rate of 7–23% reported in the IMRT-SIB dose-escalation meta-analysis studies [21] and compares well with the rate of 25–36% reported for preoperative CRT with a conventional RT dose of 50.4 Gy [18,37]. These data demonstrate the feasibility of our intensified preoperative CRT program in the combined modality treatment of LARC. Importantly, the low rate (3.5%) of major complications in patients who underwent LE compares well with the results of previous phase II trials using conventional preoperative 3D CRT and CB or CT intensification [5,6,27]; this supports the feasibility and safety of an organ-preserving approach with LE also following this IMRT-SIB dose-escalation program. Our LR rate of 9.2% appears less favorable when compared to the local control rate of 92.6% (LR = 7.4%) reported in the INTERACT trial in the dose-escalation arm with CB 3D-CRT up to 55 Gy, and to those reported from historical preoperative FU-based CRT +/− oxaliplatin trials, which ranged from 7.1 to 8.68% [13,15,37]. However, given the proportion of high-risk patients with more distant tumors, T4 or T3 MRF involvement, and clinically N2-positive nodes in our series, our local control rate remains of interest.

Importantly, the LR rate of high-risk patients was 10%, similar to the 8% rate of low and intermediate-risk patients in the overall patient population, indicating the efficacy of our intensified preoperative CRT program. The DM rate of 21.3% is consistent with the rates reported in the previously mentioned phase III trials, which ranged from 19.2% to 29.8% [15,17,37]. Although some of the patients included in these trials as well as in our retrospective study received adjuvant FU-based chemotherapy +/− oxaliplatin, DM remains the main cause of failure in all these studies. In particular, the DM rate was higher, marginally significant, in the high-risk sub-set of patients reaching a 24.5% compared to 16.2% of low and intermediate risk (*p* = 0.062). This discrepancy in DM incidence suggests that our IMRT-SIB and capecitabine treatment may have differing efficacy in different risk groups. These data are consistent with the results of the pooled data analysis of the large European trials by Valentini et al. [38], who characterized the different risk categories for outcome prediction of LARC. Our 5-year PFS and OS rates of 72.3% and 85.9% are comparable to the PFS and OS rates of the mentioned phase III trials, which are in the range of 64–73.8% vs. 69–70.5% and 79–83% vs. 80–81.3% for the control and treatment arms, respectively, emphasizing the need for a more effective systemic component of treatment. Importantly, our unfavorable results for PFS in high-risk patients compared to low and intermediate-risk (68.4% and 78.6%, *p* = 0.039) and for OS (81.2% vs. 93.7%, *p* = 0.026) support this indication, particularly for high-risk patients [38]. Some parameters such as cT4 disease and poor TRG3–5 were associated with worse PFS (*p* = 0.046), OS (*p* = 0.039) and increased incidence of DM (*p* = 0.019) as reported in our multivariable analysis characterizing this high-risk subset of patients with poor prognosis.

The recent achievements with the TNT strategy moving the systemic therapy component earlier, prior to surgery, are gaining a growing interest [39,40,41]. The optimal integration of an effective RT component, as IMRT-SIB, with more active chemotherapy, including further refinements on dose escalation, and the selection of patients for this strategy are currently under investigation and represent an important area of research in the management of LARC at present.

The unfavorable association of adjuvant chemotherapy with PFS as well as the unfavorable trend for DM incidence in multivariable analysis should be interpreted with caution due to the small number of patients treated and the possible confounder by treatment indication. No parameters associated with worsened LR were found in our patient cohort. These results confirm the efficacy of our IMRT-SIB program with moderate dose escalation. Indeed, an IMRT-SIB dose ≥ 50 Gy was associated with a favorable trend in the major response TRG1-2 (*p* = 0.056).

The efficacy of IMRT-SIB and concurrent capecitabine on local tumor control, combined with the reported good tolerability, promoted our interest in including this intensified treatment in the ongoing multicenter phase II trial of TNT, which is investigating the impact on disease control and survival in high-risk patients [42], and in the multicenter phase III trial investigating the impact of extending the time interval to surgery on tumor response after IMRT-SIB and capecitabine on organ preserving strategy in low and intermediate-risk patients [43].

Our study has some limitations. The retrospective design introduces a potential bias in the data analysis, and the different SIB doses used in clinical practice affect the homogeneity of treatment, although most patients (>80%) received a dose of 54–55 Gy. In addition, no centralized review of treatment planning was provided. Although all centers participated in the AIRO-supported national survey on GTV definition, dose prescription and treatment delivery for dose intensification in LARC and shared the defined operational criteria, these aspects may limit the generalizability of our findings to broader clinical practice. Moreover, we were unable to report on late toxicity at the long-term follow-up (only a few centers have currently submitted evaluable data). This is a critical point in dose escalation with IMRT-SIB because of the risk of severe late morbidity [44]. Data on late toxicity and long-term rectal function could be the subject of a further report. Despite these limitations, our experience with this innovative and increasingly interesting preoperative CRT intensification of LARC is reported from a multicenter, real-life clinical practice, which is a major strength of the study.

## 5. Conclusions

Preoperative IMRT-SIB and concurrent administration of capecitabine were shown to be feasible and effective in the real-life clinical practice of our multicenter study. The use of moderate dose intensification in the range of 52.5–57.5 (median 55 Gy) with the full dose of concurrent capecitabine resulted in a very low rate of grade 3 acute toxicity, high compliance to treatment with high pCR, pT0-TRG1 and tumor and nodal downstaging rates. An organ-preserving LE or W-W strategy was shown to be feasible in carefully selected, fully responding patients with a promising long-term rectal preservation rate. Despite the remarkable number of high-risk patients, the favorable long-term results in terms of LR, DM, PFS and OS highlight the efficacy of this intensified preoperative treatment compared to randomized trials using a more conventional CRT component. Given the high incidence of DM in high-risk patients compared to low and intermediate-risk patients, the incorporation of IMRT-SIB and capecitabine in a more effective systemic treatment strategy as TNT for high-risk patients focused on disease control and survival, and the use of IMRT-SIB in low-intermediate-risk patients in a strategy mainly oriented to organ preservation, has become a new area of our investigational interest.

## Figures and Tables

**Figure 1 cancers-15-05702-f001:**
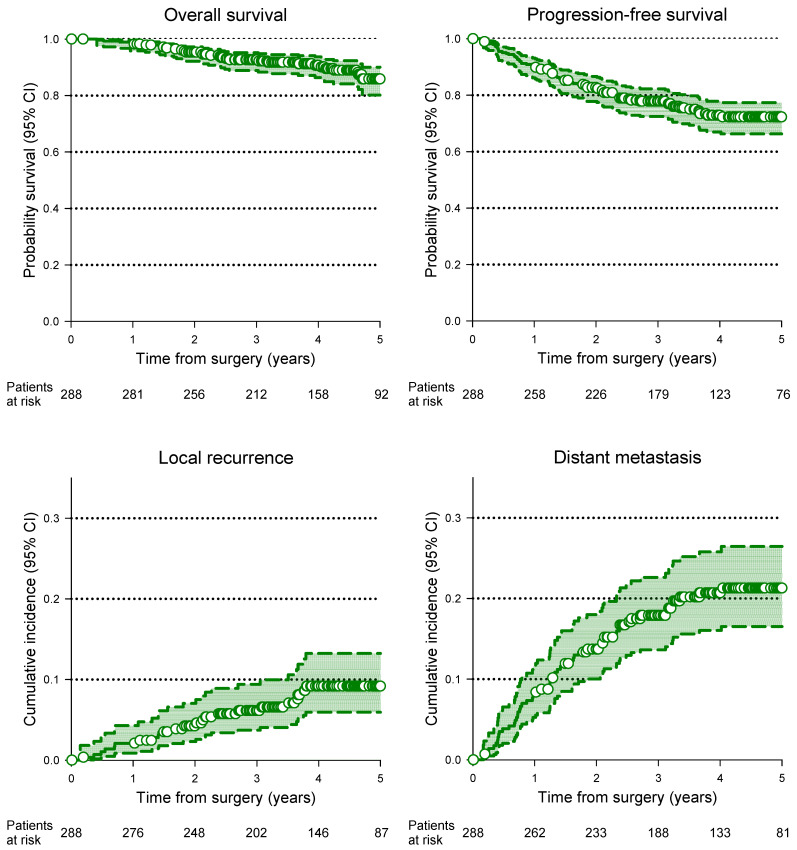
Kaplan-Meier estimates of overall survival, progression-free survival, local recurrence-free survival, and distant metastasis- free survival of the entire patient population.

**Figure 2 cancers-15-05702-f002:**
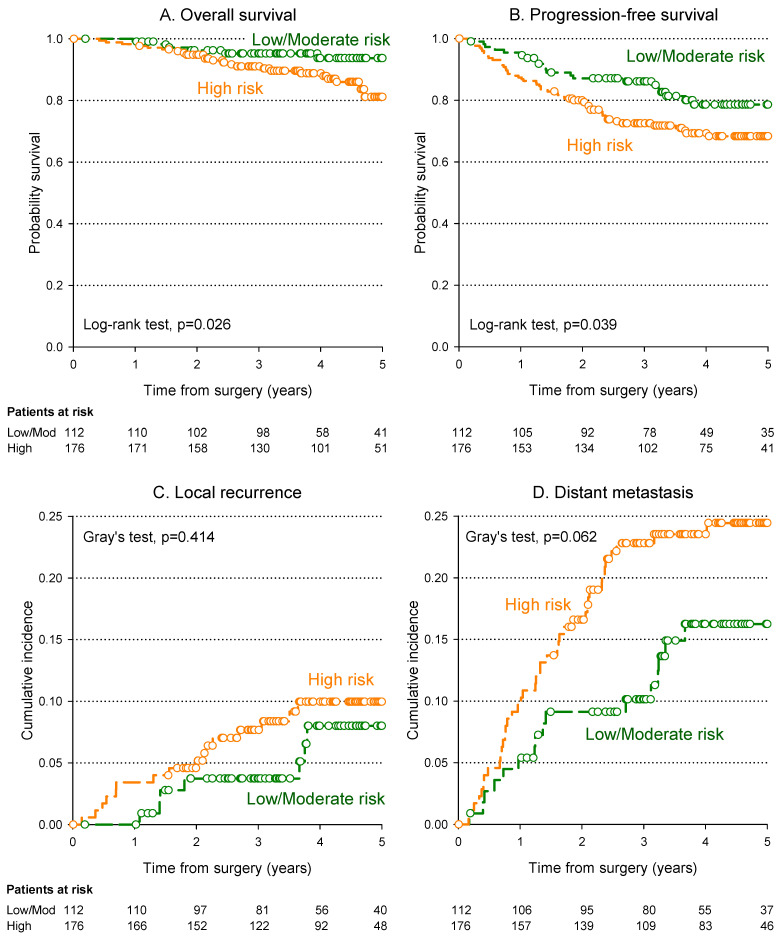
Kaplan–Meier estimates of overall survival (**A**), progression-free survival (**B**), cumulative incidence of local recurrence (**C**) and distant metastasis (**D**) according to patient risk category.

**Table 1 cancers-15-05702-t001:** Patient and Tumor Characteristics.

Characteristics	N. of Patients *n* = 288	%
**Age (years)**		
Median (Q1–Q3)	64 (55–73)	
**Gender**		
Male	184	63.9
Female	104	36.1
**Performance Status ECOG**		
0	257	89.6
1–2	31	10.2
**Tumor Location**		
Proximal Rectum	-	-
Middle Rectum	133	46.2
Distal Rectum	153	53.8
**Distance from AV** (mm)		
Median (Q1–Q3)	50 (30–75)	
**Clinical tumor stage**		
T1	-	-
T2	19	6.6
T3	215	74.7
T4	54	18.8
**Clinical nodal stage**		
N0	67	23.3
N1	130	45.1
N2	91	31.6
**MRF involvement**		
No	145	50.4
Yes	139	48.3
Unknown	4	1.4
**Risk category**		
Low/moderate	112	38.9
High	176	61.1
**Tumor Histology**		
Grade1 Adenocarcinoma	16	5.5
Grade2 Adenocarcinoma	245	85.0
Grade3 Adenocarcinoma	20	7.0
Unknown	7	2.4

AV: Anal Verge; MRF: Mesorectal Fascia.

**Table 2 cancers-15-05702-t002:** Preoperative Chemoradiation, Clinical Response, Acute Toxicity and Treatment Compliance.

	No of Patients *n* = 288	%
**IMRT-SIB dose (dose fraction)**		
52.5 Gy (2.1 Gy)	23	8.0
54 Gy (2.16 Gy)	91	31.6
55 Gy (2.2 Gy)	141	49.0
56 Gy (2.25 Gy)	22	7.6
57.5 Gy (2.3 Gy)	11	3.8
**Interruption/Delayed IMRT**		
No	286	99.3
Yes	2	0.7
**Interruption/delayed Capecitabine**		
No	225	78.1
Yes	63	21.9
**Clinical Response**		
(patients selected for organ preservation)		
cCr *	20	6.9
mCR **	16	5.6
**Acute toxicity**		
Grade 3–4 gastrointestinal	17	5.9
Grade 3–4 hematologic	6	2.1
Grade 3–4 urologic	0	0.0
Grade 3–4 other	3	1.0
**Treatment Compliance**		
IMRT-SIB Concurrent capecitabine	286 225	99.3 78.1

** mCR: majorClinical Response; * cCR: completeClinical Response.

**Table 3 cancers-15-05702-t003:** Correspondence between clinical and pathological T stage in patients that underwent Local Excision.

Clinical Response		ypT Stage	
	pT0 (*n* = 19)	* pT1 (*n* = 6)	** pT > 1 (*n* = 4)
Complete (*n* = 13)	11 (84.6%)	2 (15.4%)	0 (0.0)
Major (*n* = 16)	8 (50.0%)	4 (25.0%)	4 (25.0%)

* 2 of 4 pts had completion of a TME(TRG3); ** all 4 pts had completion of a TME.

**Table 4 cancers-15-05702-t004:** Surgery, pathological findings, non-operative management and postoperative complications.

Variables	N. of Patients (*n* = 288)	%
**Surgical procedure**		
LAR	187	64.9
APR	62	21.5
Hartmann’s resection	3	1.0
Local Excision	29	10.1
Non-Operative Management	7	2.4
**Tumor Pathological Stage**		
Total mesorectal excision (*n* = 252)		
pT0N0 (pCR)	65	25.8
pT0N1	4	1.6
pT1	22	8.7
pT2	75	29.8
pT3	84	33.3
pT4	2	0.8
Local excision (*n* = 29)		
pT0Nx *	19	65.5
pT1Nx **	6	20.7
pT2Nx/pT3Nx	4	13.8
**Mandard TRG**		
TRG1	85	29.5
TRG2	92	31.9
TRG3	68	23.6
TRG4–5	22	7.6
Unknown	21	7.3
**Nodal pathological stage** (among patients undergoing TME)		
pN0	201	79.8
pN1	42	16.7
pN2	9	3.6
**CRM**		
>1 mm	238	94.5
≤1 mm	14	5.5
**Postoperative Complications**		
Leakage anastomosis (LAR)	15	8.0
Pelvic abscess	8	3.2
Fistula	4	1.4
Perineal wound infection (APR)	3	4.8
Small bowel (sub-)obstruction	4	1.4
Ileus	4	1.4
Other	14	5.0

* patients with TRG1 or ** TRG2 after local excision; rates evaluated on all 279 operated pts (TME + LE); multiple occurrences Clavien–Dindo ≥ 3. Legend: TRG: tumor regression grade; CRM: circumferential resection margin.

**Table 5 cancers-15-05702-t005:** Patterns of failure according to surgical options following preoperative CRT.

Surgery	Patients	Outcome, *n* (%)				
		LR	LR + DM	DM	Death	Disease Free
TME	252	10 (4.0)	10 (4.0)	44 (17.4)	30 (11.9)	184 (73.0)
LE *	29	1 (3.4)	1 (3.4)	2 (6.9)	1 (3.4)	26 (86.2)
W-W **	7	1 (14.3)	0 (0.0)	0 (0.0)	0 (0.0)	6 (85.7)

LR local recurrence; DM distant metastasis; * 6 LE pts had the completion of TME; ** 2 regrowth pts had TME or LE; 2 pts had salvage TME.

**Table 6 cancers-15-05702-t006:** Multivariable odds ratio (OR) and corresponding 95% confidence interval (CI) ^a^ for TRG3-5 vas TRG1-2 according to clinical characteristics.

Variables	Mandard TRG3-5 vs. TRG1-2	
	OR (95% CI)	*p*-Value
Age ≥ 65 years	0.67 (0.40–1.14)	0.137
Male gender	1.73 (0.97–3.06)	0.061
cT4 vs. cT2-cT3	0.96 (0.44–2.06)	0.907
cN2 vs. cN0-cN1	1.02 (0.58–1.79)	0.950
Positive MRF involvement	2.00 (1.05–3.79)	0.035
Distance from anal verge ≥5 cm	1.37 (0.79–2.39)	0.266
Time to surgery >10 vs. ≤10 months ^b^	1.35 (0.58–3.13)	0.491
Planned IMRT-SIB do	0.57 (0.32–1.01)	0.056

^a^ Estimated from unconditional logistic regression model including all variables in the table. ^b^ From the end of radiotherapy.

**Table 7 cancers-15-05702-t007:** Multivariable hazard ratios (HRs) and corresponding 95% confidence intervals (CIs) ^a^ for overall and progression-free survival.

Variables	Overall Survival		Progression-Free Survival	
	HR (95% CI)	*p*-Value	HR (95% CI)	*p*-Value
Age ≥ 65 years	1.82 (0.86–3.82)	0.116	0.88 (0.55–1.41)	0.589
Male gender	1.90 (0.78–4.62)	0.157	1.58 (0.93–2.67)	0.091
cT4 vs. cT2-cT3	2.61 (1.05–6.49)	0.039	1.97 (1.01–3.84)	0.046
cN2 vs. cN0-cN1	0.73 (0.33–1.63)	0.444	0.82 (0.49–1.39)	0.465
Positive MRF involvement	1.51 (0.58–3.95)	0.398	0.97 (0.53–1.77)	0.909
Distance from anal verge ≥ 5 cm	1.35 (0.62–2.98)	0.452	0.84 (0.51–1.37)	0.482
Time to surgery >10 vs. ≤10 months ^b^	0.46 (0.06–3.43)	0.446	1.04 (0.47–2.32)	0.915
Mandard TRG3–5 vs. TRG1-2	2.62 (1.09–6.30)	0.031	1.75 (1.03–3.00)	0.040
pT0N0 vs. other	0.86 (0.29–2.56)	0.785	1.79 (0.81–3.95)	0.152
IMRT-SIB dose ≥ 55 Gy	1.26 (0.55–2.87)	0.585	0.89 (0.53–1.48)	0.647
Adjuvant chemotherapy	1.25 (0.56–2.76)	0.588	1.91 (1.15–3.18)	0.013

^a^ Estimated from the Cox proportional hazard model including all variables in the table. ^b^ From the end of radiotherapy.

**Table 8 cancers-15-05702-t008:** Multivariable hazard ratios (HRs) and corresponding 95% confidence intervals (CIs) ^a^ for local recurrence and distant metastasis.

Variables	Local Recurrence		Distant Metastasis	
	HR (95% CI)	*p*-Value	HR (95% CI)	*p*-Value
Age ≥ 65 years	0.72 (0.28–1.85)	0.499	0.89 (0.53–1.49)	0.652
Male gender	2.85 (0.97–8.40)	0.057	1.29 (0.74–2.25)	0.364
cT4 vs. cT2-cT3	0.91 (0.27–3.08)	0.879	2.17 (1.07–4.39)	0.032
cN2 vs. cN0-cN1	0.99 (0.36–2.77)	0.990	0.76 (0.43–1.35)	0.349
Positive MRF involvement	1.06 (0.36–3.12)	0.923	0.97 (0.50–1.87)	0.919
Distance from anal verge ≥5 cm	0.58 (0.26–1.31)	0.192	0.88 (0.51–1.49)	0.622
Time to surgery >10 vs. ≤10 months ^b^	1.44 (0.34–6.12)	0.621	0.76 (0.30–1.94)	0.571
Mandard TRG3–5 vs. TRG1-2	1.15 (0.43–3.12)	0.779	1.99 (1.12–3.54)	0.019
pT0N0 vs. other	1.47 (0.39–5.57)	0.572	2.14 (0.87–5.29)	0.098
IMRT-SIB dose ≥55 Gy	0.93 (0.39–2.19)	0.863	0.82 (0.48–1.43)	0.491
Adjuvant chemotherapy	2.30 (0.86–6.21)	0.099	1.71 (0.99–2.97)	0.055

^a^ Estimated from the Cox proportional hazard model including all variables in the table and adjusted for competing risk according to the Fine–Gray model. ^b^ From the end of radiotherapy.

## Data Availability

Data can be supplied upon reasonable request with the corresponding author.

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
