# Peer review of "Preoperative Intensified Chemoradiation with Intensity-Modulated Radiotherapy and Simultaneous Integrated Boost Combined with Capecitabine in Locally Advanced Rectal Cancer: Long-Term Outcomes of a Real-Life Multicenter Study"

_cancers, 2023, doi:10.3390/cancers15235702_

Round 1

Reviewer 1 Report

Comments and Suggestions for Authors

While the article presents a comprehensive analysis of a cohort of locally advanced rectal cancer (LARC) patients treated with preoperative intensity-modulated RT and simultaneous boost (IMRT-SIB) along with capecitabine, there are a few potential downsides to consider:

Limited Generalizability: The study is retrospective and based on data from 10 Italian Institutions. This may limit the generalizability of the findings to a broader population or different healthcare settings and this aspect needs to be mentioned under Study Limitations.

Selection Bias for Organ Preservation: The article mentions that 36 patients (12.5%) with complete or major clinical response were offered an organ preservation approach. It would be helpful to understand how these patients were selected, as this could introduce a potential selection bias. Imaging / clinical? criteria for this aspect needs to be explained.

Postoperative Complications: The article reports a 14.2% incidence of major postoperative complications. This is a relatively high percentage and could be a concern for the overall well-being and recovery of the patients. A Clavien-Dindo approach to complications may aid in the overall strength of the article.

Distant Metastasis Rate: The 21.3% rate of distant metastasis at a median follow-up of 50 months is noteworthy. It suggests that despite the treatment, there is still a significant risk of distant recurrence, which could be an area for further investigation or improvement in the treatment approach.

High-Risk Subset: The article highlights a higher distant metastasis rate (24.5%) in the high-risk subset compared to the low-intermediate risk group (16.2%). This discrepancy suggests that the treatment may have differing efficacy in different risk groups, which could impact its applicability and needs to be further explained.

Potential for Investigational Interest: While the article suggests that the incorporation of IMRT-SIB with more effective systemic therapy in high-risk patients and its use in organ preservation for low-intermediate risk patients may be areas of investigational interest, this also implies that further research is needed to optimize the treatment approach.

Overall, the study provides valuable insights into the long-term outcomes of preoperative IMRT-SIB and capecitabine in LARC patients. However, it's important to acknowledge these potential downsides to fully understand the implications and consider areas for future research or refinement of the treatment protocol.

Comments on the Quality of English Language

The abbreviation pts instead of patient / patients is used to often in the text leading to an unpleasant reading experience (Line: 324, 180, 173 etc..)

Author Response

Author response to report 1:

1st Comment: Limited Generalizability: The study is retrospective and based on data from 10 Italian Institutions. This may limit the generalizability of the findings to a broader population or different healthcare settings and this aspect needs to be mentioned under Study Limitations.

Reply to the 1st Comment: Thanks for this remark. We agree that the retrospective design of the study has some limitations in the data analysis and homogeneity of treatment. We added also the limited generalizability in the Study Limitations as you suggested “…, these aspects may limit the generalizability of our findings to broader clinical practice”, line 517-518.

2nd Comment: Selection Bias for Organ Preservation: The article mentions that 36 patients (12.5%) with complete or major clinical response were offered an organ preservation approach. It would be helpful to understand how these patients were selected, as this could introduce a potential selection bias. Imaging / clinical? criteria for this aspect needs to be explained.

Reply to 2nd Comment: Thanks tor this comment. As we reported in Mat and Methods, Clinical Response was assessed at restaging with pelvic MRI, DRE, endoscopy and endorectal ultrasound (EUS), if available and graded in cCR and mCR (lines 196-204). Indeed, the selection of patients for organ preservation was based on the enrollment in phase II organ preservation trials and in some patients was based on individual preference, surgeon attitude and the institution's multidisciplinary team experience (lines 433-438 in Discussion); we agree that potential bias exist, but these may be inherent to institution evolving experience as reported in lines 440-441.

3rd Comment. Postoperative Complications: The article reports a 14.2% incidence of major postoperative complications. This is a relatively high percentage and could be a concern for the overall well-being and recovery of the patients. A Clavien-Dindo approach to complications may aid in the overall strength of the article.

Reply to 3rd Comment: Thanks for this remark. We used the Clavien- Dindo classification to evaluate postoperative complications (lines 220-221 Mat-Methods) and our major complication rate of 14.2% resulted consistent with data reported in the IMRT-SIB dose-escalation meta-analysis studies and compares favorably  with the rate of 25-36% reported from CRT trials using conventional RT, as reported in our Discussion section (lines 457-459)

4th Comment.  Distant Metastasis Rate: The 21.3% rate of distant metastasis at a median follow-up of 50 months is noteworthy. It suggests that despite the treatment, there is still a significant risk of distant recurrence, which could be an area for further investigation or improvement in the treatment approach.

Reply to 4th Comment: Thanks for this comment. We agree that the incidence of DM remain the major problem. Indeed, the systemic component of our IMRT-SIB is concurrent capecitabine administration only and the impact of adjuvant chemotherapy confirmed uncertain also in our experience. The implications for further investigations are reported below in reply of the other comments.

5th Comment. High-Risk Subset: The article highlights a higher distant metastasis rate (24.5%) in the high-risk subset compared to the low-intermediate risk group (16.2%). This discrepancy suggests that the treatment may have differing efficacy in different risk groups, which could impact its applicability and needs to be further explained.

Reply to 5th Comment: Thanks for addressing this important point. While the low incidence of local recurrences was similar in the two different risk groups, the higher incidence of DM and the unfavorable results for PFS and OS in high-risk patients compared to low and intermediate-risk support the need of more effective systemic therapy component, particularly for high-risk group as we reported in the discussion section. Indeed, IMRT-SIB and capecitabine was mainly effective on local tumor control rather than on its metastatic potential. We added “This discrepancy in DM incidence suggests that our IMRT-SIB and capecitabine treatment may have differing efficacy in different risk group” in lines 480-481. Further explanation is reported in lines 485-488. This is further specified adding this consideration in line 504:         “The efficacy of IMRT-SIB and concurrent capecitabine on local tumor control, combined with” the reported good tolerability,  in promoting our interest to include this intensified treatment in the ongoing multicenter phase II TNT study in high-risk patients and to investigate its application in organ preservation strategy in a phase III trial for low-intermediate risk patients.

6th Comment. Potential for Investigational Interest: While the article suggests that the incorporation of IMRT-SIB with more effective systemic therapy in high-risk patients and its use in organ preservation for low-intermediate risk patients may be areas of investigational interest, this also implies that further research is needed to optimize the treatment approach.

Reply to 6th Comment: Thanks for these remarks. We agree that treatment optimization including refinements of IMRT-SIB dose escalation could be an area of further investigation. We added this consideration on lines 494-498 “The optimal integration of an effective RT component, as IMRT-SIB, with more active chemotherapy, including further refinements on dose escalation, and the selection of patients for this strategy …”

Comments on the Quality of English Language: We made an English revision of the text and delated the patient abbreviation (pts) according to your recommendation.

Reviewer 2 Report

Comments and Suggestions for Authors

In this study, the authors analyzed the long-term results of a retrospective multicentric experience on preoperative capecitabine-based CRT intensification with IMRT-SIB in real-life clinical practice at 10 Italian institutions.  They found that the use of moderate IMRT-SIB dose intensification with full-dose of concurrent capecitabine was safe and well tolerate. The conclusions were partly supported by solid data. However, a few issues need to be addressed.

1.       The quality of presentation is in a mess and there are lots of mistakes, the authors should review the paper.

2.       In Line 325, it should be Table 5 instead of Table 4?

        3. Lack of control in this study?

        4. The response was poor in high-risk group may also be due to the pts were in high tumor stage?

        5. What is the promising application of this study? 

Comments on the Quality of English Language

Must be improved

Author Response

Author response to Reviewer 2

1st Comment: he quality of presentation is in a mess and there are lots of mistakes, the authors should review the paper.

Reply to the 1st Comment: Thanks for this comment. We carefully reviewed the manuscript. Indeed, we removed several mistakes of the text. We hope the presentation now to be improved

2nd Comment: In Line 325, it should be Table 5 instead of Table 4?

Reply to the 2nd Comment: thanks for this comment. We confirm this is the Table 4: Surgery, pathological findings, non-operative management, and postoperative complications, as reported. In Table 5 are reported: Patterns of failure according to surgical options following preoperative CRT.

3rd Comment: Lack of control in this study?

Reply to 3rd Comment: thanks for this remark. As we reported in “Study design and objectives” this is a real-life retrospective study of a cohort of 288 patients with LARC collected in a prospective database of 10 Italian institutions. Quality control of data was made at each participating institution.

4th Comment: The response was poor in high-risk group may also be due to the pts were in high tumor stage?

Reply to 4th comment: Thanks for this comment. As the high-risk group include poor prognostic factors, as T4 stage, MRF involvement and or N2 disease, the poor outcomes reported in these patients, in particular the DM, PFS and OS may be correlated with the more advanced disease, as confirmed at our multivariable analysis.

5th Comment: What is the promising application of this study? 

Reply to 5th Comment: Thanks for addressing this important point. The efficacy of IMRT-SIB and concurrent capecitabine on local tumor control combined with the reported good tolerability may represent an appealing treatment strategy in LARC. As we reported in the discussion section, the results of the study promoted  our investigational interest including this intensified treatment in the ongoing multicenter phase II trial of TNT in high-risk patients and in phase III trial investigating the impact of extending the time interval to surgery on tumor response after IMRT-SIB and capecitabine on organ preserving strategy in low and intermediate-risk patients.

Comments on the Quality of English Language: We made an English revision of the text  according to your recommendation

Round 2

Reviewer 2 Report

Comments and Suggestions for Authors

This is a promising clinical data for the future application, and the authors answered all the issues that were raised.